# Image Forensics Using Non-Reducing Convolutional Neural Network for Consecutive Dual Operators

**Se-Hyun Cho [1], Saurabh Agarwal [2,3], Seok-Joo Koh [1,* and Ki-Hyun Jung [3,***

[1] School of Computer Science and Engineering, Kyungpook National University, Daegu 41566, Korea; csh831219@knu.ac.kr
[2] School of Engineering & Technology, Amity University, Noida 201301, Uttar Pradesh, India; sagarwal7@amity.edu
[3] Department of Cyber Security, Kyungil University, Gyeongsan 38424, Korea
* Correspondence: sjkoh@knu.ac.kr (S.-J.K.); kingjung@kiu.kr (K.-H.J.)

**Abstract:** Digital image forensics has become necessary as an emerging technology. Images can be adulterated effortlessly using image tools. The latest techniques are available to detect whether an image is adulterated by a particular operator. Most of the existing techniques are suitable for high resolution and manipulated images by a single operator. In a real scenario, multiple operators are applied to manipulate the image many times. In this paper, a robust moderate-sized convolutional neural network is proposed to identify manipulation operators and also the operator's sequence for two operators in particular. The proposed bottleneck approach is used to make the network deeper and reduce the computational cost. Only one pooling layer, called a global averaging pooling layer, is utilized to retain the maximum flow of information and to avoid the overfitting issue between the layers. The proposed network is also robust against low resolution and JPEG compressed images. Even though the detection of the operator is challenging due to the limited availability of statistical information in low resolution and JPEG compressed images, the proposed model can also detect an operator with different parameters and compression quality factors that are not considered in training.

**Keywords:** image forensics; image operator sequence; convolutional neural network; image manipulation; image forgery detection; deep learning technique

## 1. Introduction

Digital images are a victim of manipulation due to the ease of availability of high precision but uncomplicated image editing tools. Image forensics is needed to detect the source, processing history, and genuineness of the image. Numerous methods [1–3] are provided to find the source device of the image. The mismatch of source assists in image forgery detection as a fake image is created usually by using two or more images. Most of the fake images look realistic by applying multiple spatial operations. The detection of operations such as resampling [4,5], sharpening [6–8], and median filtering [9,10] can make uncovering the image forgery easy. Many universal methods [11–21] also exist to detect the image forgery operations simultaneously. However, universal methods can only effectively detect a single operation on an image. In the real scenario, more than one operation is applied to the image in general. In this paper, a sequence of operations can be detected perfectly to unfold the processing history of the image. Few techniques [22–26] exist that can detect the operations and the order of operations. Although the performance is not consistent according to different operations, JPEG compression has a significant role in the forensic analysis as the most common format. The performance in most of the existing techniques degrades while considering JPEG compression.

In the recent era of a deep learning network, a convolutional neural network has given propitious results in many applications. A convolutional neural network (CNN) is utilized in the detection of median filtering, resampling, universal image manipulation, multiple

JPEG compression, contrast enhancement, image splicing, etc. Qiu et al. [11] discovered that some existing techniques, especially LBP [27] and SRM [28], can effectively detect different types of image operations such as Gaussian filtering, median filtering, image resizing, gamma correction, and compression history. The experiments are performed on large-size images, where the experimental analysis is limited as different compression qualities and filter sizes are combined in one data set. Bayar and Stamm [12] introduced the CNN for the detection of additive white Gaussian noise, Gaussian filtering, median filtering, and image resizing for the first time. In particular, a constrained design is applied in the first layer of the proposed CNN. Experimental results are given for $227 \times 227$ image blocks. However, no experimental analysis was given for low resolution and JPEG compressed images. The proposed idea of constraint is further extended with an improved CNN model [13], and JPEG compression is used in the experimental analysis. In the improved CNN architecture, a constrained convolutional layer is followed by four blocks, and each block contains a convolutional layer, batch normalization layer, ReLU layer, and pooling layer. Further, three softmax classification layers are used, and the outputs are classified using an extremely randomized tree classifier. The constrained convolutional layer filters can predict the errors by subtracting the resultant value from the central value of the filter window. The constraint is enforced during the training of each iteration. However, the performance of the improved CNN model falls in the most cases when two operators are applied consecutively on the image even on a large size image. Li et al. [14] selected some sub-models from SRM by calculating the out-of-bag error. The selection process can reduce the feature dimension noticeably. The results are analyzed for eleven image operations of several categories like spatial filtering, image enhancement, and JPEG compression. The proposed technique also claims good results in the detection of four anti-forensic operations such as JPEG compression, contrast enhancement, resampling, and median filtering. However, the performance degrades for small-size images. Boroumand and Fridrich [15] proposed a model using CNN and multilayer perceptron (MLP) for high-pass filtering, low-pass filtering, de-noising, and tonal adjustment that has four types of operations. Eight convolutional layers are utilized in the proposed CNN. The MLP classifies the images using moments that are extracted in the last part of the CNN model. The aforementioned method is also compared with the manual feature extraction method. The experiments are discussed for $512 \times 512$ size images only. Mazumdar et al. [16,17] utilized the Siamese network for pair-wise learning. Two identical networks are used to classify multiple operations—median filtering, Gaussian filtering, gamma correction, additive white Gaussian noise, and image resizing. The authors claim that the proposed method can detect a processed image for an unseen operation that is not included in the training. The choice of a two-stream CNN architecture gives better performance than analogous single-stream architecture [13]. However, the model capability decreases on unknown data sets abruptly. Chen et al. [18] discussed the densely connected CNN model for detecting eleven types of operations. Each dense block is followed by a transition layer and pooling layer. The transition layer performs $1 \times 1$ convolutions to reduce the number of feature maps and computation costs. The dataset of training and testing is the same which can make unbiased performance evaluation difficult. Xue et al. [19] applied the Siamese network for identifying some operations like as inclusion of text, logo, and black block in the image. Operations also include image resampling, Gaussian noise, and Gamma correction. The Siamese network utilized the AlexNet and ResNet-18. Uncompressed images are considered only in experiments. Singhal et al. [20] introduced a CNN model with two convolutional layers only to detect seven types of operations. DCT coefficients of median filter residual are used as input in the CNN network. Large-size filters are used in the convolutional layer. Barni et al. [21] detected image operations like median filtering, image resizing, and histogram equalization. The features are extracted using two neural networks [12,29]. A random feature selection approach is utilized to select the robust features from the CNN network, and an SVM classifier is applied to find the type of attack finally.

Detection of image operations order is also a great concern for a deep understanding of image processing history. Some efforts [22–26] are performed to detect the order of operations. In [22,23], an analysis is given to find the reasons for the non-detection of operator sequence order, where a framework based on mutual information is suggested. The methods are not able to detect some operator sequences. JPEG compressed images are not detectable using the previous model. Comesaña [24] discussed the theoretical possibilities of operator order detection. Bayar and Stamm [25] discussed a CNN with a constrained convolution layer for estimating the order of operations. Liao et al. [26] suggested two-stream CNN, to find out the operators and their respective orders. Though a customized preprocessing is required to apply for each operation, the method can detect operation even for an unknown parameter on the same operation using weight transfer.

In this paper, a non-reducing convolutional neural network is proposed that can assure the maximal flow of details between layers. The specific contributions of the proposed network can be outlined as follows:

- The proposed non-reducing CNN can detect a dual-operated image and the operation sequence. Different types of operations such as median filtering, Gaussian blurring, image resizing, and un-sharp masking are detected successfully.
- Multiple convolutional layers are inserted in the CNN network by adopting a bottleneck approach in the proposed method. The computational requirement of the proposed CNN is less due to fewer learning parameters. However, the proposed method has a performance improvement by the bottleneck approach.
- To retain maximum statistical information, no pooling layer is interleaved between the convolutional layers. Since a pooling layer can reduce the computational cost with the sacrifice of relevant operation fingerprints that are inherited.
- To avoid the overfitting issue and boost the performance, one global averaging pooling layer is utilized. An additional improvement of more than two percent in the detection accuracy can achieve by using a global averaging pooling layer in most of the cases.
- The proposed method can ensure a better performance in challenging environments with low-resolution images and dual operators manipulation without specific preprocessing requirements.

The remaining paper is organized as follows. In Section 2, a problem is formulated for the dual operator manipulation detection in multiple scenarios. The proposed non-reducing CNN model is explained in Section 3. Detailed experimental analysis is performed in Section 4 with comparative analysis. The important advantages of the proposed scheme are highlighted in Section 5.

## 2. Detection of Image Processing Operator Sequence

In this section, three issues are discussed to address the importance of image forensics for operator sequence. In the first, the problem of detecting image operator sequence with its order is discussed. In the second, image processing history will be discovered for compressed images. In the third, a challenging detection scenario is discussed in which specification is dissimilar.

### 2.1. Problem Formulation

Assuming that there are two operators, $\alpha$ and $\beta$, in the image operation sequence, the detection of a dual operator sequence can be understood as a multiclass classification problem. The five classes according to processing history can be formulated:

$\Omega 0$: An image is not operated by any operator;

$\Omega 1$: An image is operated by $\alpha$ operator;

$\Omega 2$: An image is operated by $\beta$ operator;

$\Omega 3$: The first image is operated by $\alpha$ then operated by $\beta$;

$\Omega 4$: The first image is operated by $\beta$ then operated by $\alpha$.

Image histograms can be visualized to recognize the detection complexity of dual operator sequence. Image histogram provides the summary of image pixels according to their intensity. The changes in the pixel intensity are unavoidable when applying any type of operator. A single image is considered to understand changes in the image after applying some operator. In Figure 1, a pristine image (ORI) with 128 × 128 pixels in BOSSbase [30] and a corresponding histogram are shown.

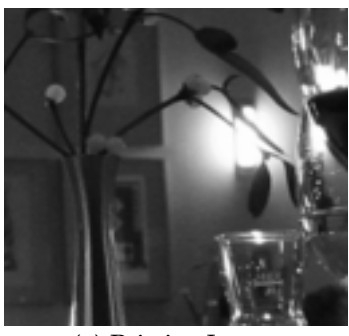

(**a**) Pristine Image                    (**b**) Histogram of pristine image

**Figure 1.** Pristine image and histogram.

In Figure 2, histograms of the pristine image (Figure 1a) are displayed after applying a single operator and dual operator sequence. Four operators—Gaussian blurring with standard deviation 1.0 (GAU_1.0), median filtering with filter size 5 × 5 (MF5), un-sharp masking sharpening with radius 3.0 (SH_3.0), up-sampling with factor 1.5 (UP_1.5) and their two operator sequence from GAU_1.0 MF5 to UP_1.5 GAU_1.0 are considered. As discussed above, five classes are possible while considering two operators—GAU_1.0 and MF5 for example. There is a slight difference in the histograms of Ω1 (GAU_1.0), Ω2 (MF5), Ω3 (GAU_1.0 MF5), and Ω4 (MF5 GAU_1.0) as shown in Figure 2. A similar pattern is also followed by other operators. The histogram of the median filtered image is very similar to the pristine image histogram. It is due to the nonlinear behavior of the median filter. Further, the internal statistical information becomes limited while operated by dual operators.

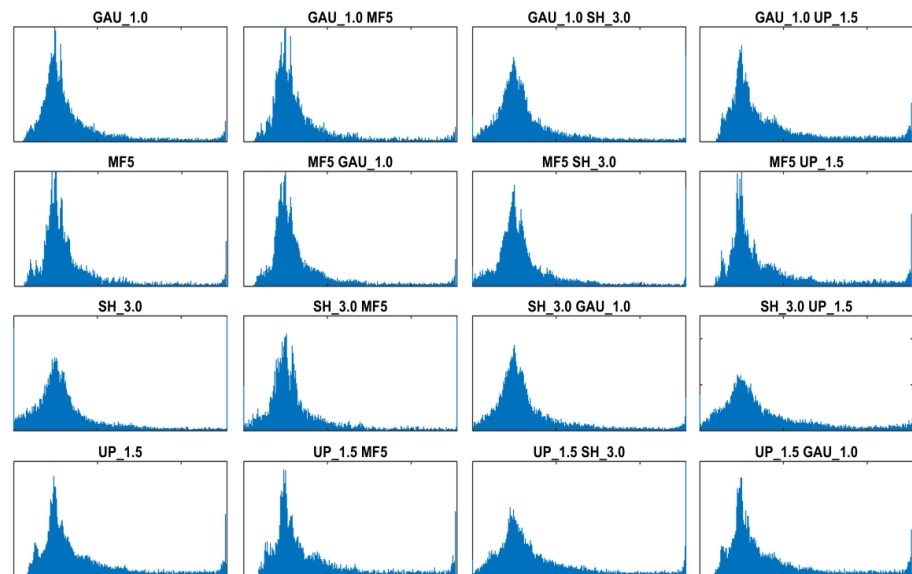

**Figure 2.** Image histogram for a single operator and dual operators.

One attempt [23] is made to detect the image resizing and Gaussian blurring operations pair. The image features are visualized in the frequency domain. However, the results are

not encouraging for the operator sequence. The Ω3 and Ω4 operators sequence are not detectable. In spite that the strength of each operator varies, one operator artifacts in dual operators sequence can be suppressed by another operator.

### 2.2. Effectiveness on Compressed Images

JPEG is a common format to store the images. Most of the digital devices are using JPEG format as default in photos. In general, a fake photo is created using JPEG images. Multiple operations are applied to create fake photos. The fake photo is required to be stored again, which brings the double JPEG compression artifacts in the fake image. It is obvious that double JPEG compression has occurred in the sequence of fake image creation. Therefore, five possible classes according to JPEG quality factors Q1 and Q2 in the operator sequence of operator α and operator β can be defined as follows:

Ω0: Image is not operated by any operator and JPEG compressed with quality factor Q1;
Ω1: Image is JPEG compressed with quality factor Q1 and operated by α operator then JPEG compressed with quality factor Q2;
Ω2: Image is JPEG compressed with quality factor Q1 and operated by β operator then JPEG compressed with quality factor Q2;
Ω3: Image is operated by α then JPEG compressed with quality factor Q1, and again the image is operated by β then JPEG compressed with quality factor Q2;
Ω4: Image is operated by β then JPEG compressed with quality factor Q1, and again the image is operated by α then JPEG compressed with quality factor Q2.

JPEG compression diminishes the artifacts of operators. Multiple operators and double JPEG compression can raise the complexity of the problem.

### 2.3. Detection for Dissimilar Parameters and Compression

In existing techniques, the operator parameters of training and testing images are the same. However, the operator parameters can be mismatched still even operator is the same in a real scenario. In [29], an attempt is proposed to detect the various type of tonal adjustments for unknown parameters. The proposed deep CNN model works efficiently for JPEG compressed images. However, the results on unknown parameters of an operator are not encouraging for using existing universal operator detectors. The constrained CNN [25] is applied for two operator sequences—Gaussian blurring and resizing. The parameters of Gaussian blurring and resizing are a standard deviation of 0.7, and a scaling factor of 1.2 for training and a standard deviation of 1.0, and a scaling factor of 1.5 for testing, respectively. The possible five classes are not classified properly, as can be seen in the confusion matrix in the two operator sequences as shown in Figure 3.

|      | Ω0   | Ω1   | Ω2   | Ω3   | Ω4   |
|------|------|------|------|------|------|
| Ω0   | 0.94 | *    | *    | *    | *    |
| Ω1   | *    | 0.61 | *    | *    | *    |
| Ω2   | *    | *    | 0.08 | *    | *    |
| Ω3   | *    | *    | *    | 0.99 |      |
| Ω4   | *    | *    | *    | *    | 0.95 |

**Figure 3.** Confusion matrix for α = GAU_0.7 and β = UP_1.5.

The requirement of robustness against dissimilar parameters is more challenging. However, it is a more practical and real situation. In this paper, a universal operator detector is proposed in a more real situation, in other words, for dissimilar parameters. The proposed deep model is suitable for both single-operator and dual-operator sequences. The

dissimilar parameters are considered in a particular range. The proposed technique can learn features for detecting operator sequences automatically by using bottleneck CNN. Bottleneck blocks required less computation cost, which can help in increasing the layers. In the proposed model, there is no need for handcrafted feature extraction and selection as required in traditional machine learning. In previous literature, customized preprocessing is needed for different operators. In the proposed techniques, images are not required to undergo any type of preprocessing. However, there is a need for preprocessing according to the operator in some previous works [20,26]. The proposed CNN can highlight the statistical anomaly and classify successfully the five classes discussed above.

## 3. Framework of the Proposed CNN

The CNN has proved its worthiness in many applications such as image classification, fake face detection, image forgery detection, etc. In this paper, a robust deep architecture is proposed to detect single and dual operators in the processed images. The proposed architecture is effective on both compressed and non-compressed images. The proposed CNN architecture can suppress the need for any preprocessing layer as used in some previous techniques [20,26]. In the previous techniques, exclusive preprocessing is required according to the operator, which is not feasible in a practical situation and restricted the network performance for particular operators. When two operations are applied simultaneously, the artifacts of the first operator can be diminished by the second operator. In Figure 4, some pairs of operators are considered to check the behavior of operations on the BOSSbase [30] image database. The normal distribution of image entropy is reflected in the plots. There is an overlap in some places for five possible cases of two operators, where the overlapping makes the problem challenging.

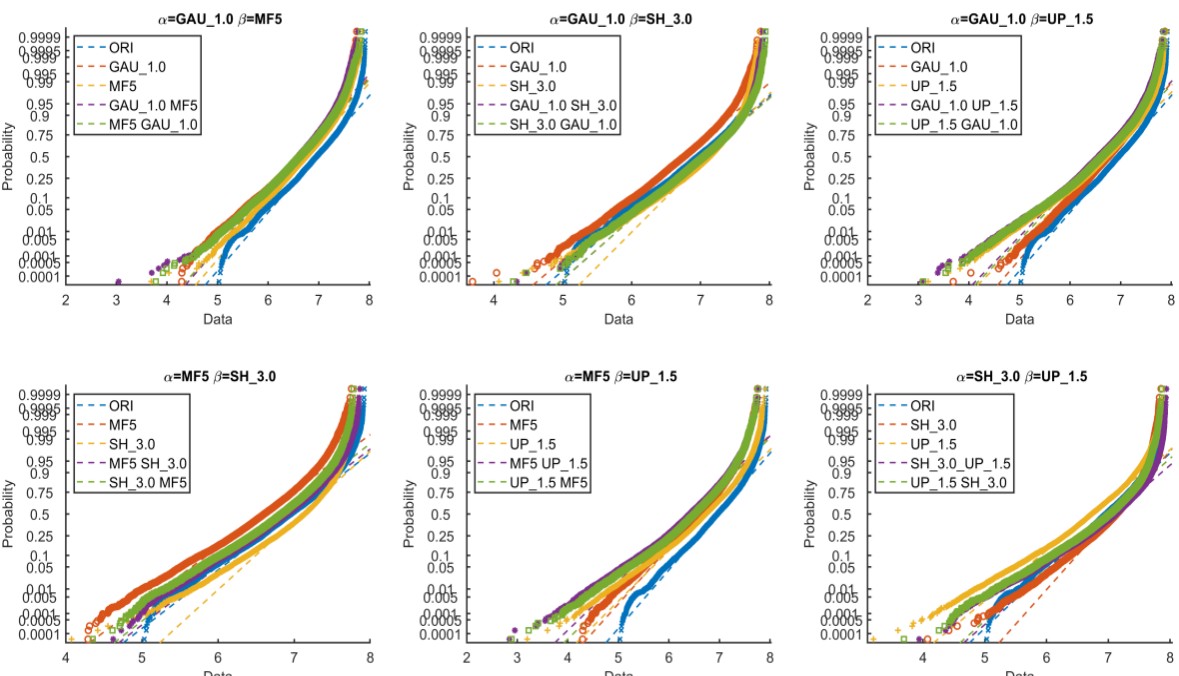

**Figure 4.** Normal distribution of entropy after applying operator on the image set.

The proposed architecture can resist the problem of operator sequence in a better way. The CNN contains multiple layers and filters to classify the input into their respective classes. The CNN parameters like weight and biases are updated as the network learns. The image input layer is followed by seven blocks and four layers in the proposed CNN. The block diagram of the proposed CNN is shown in Figure 5.

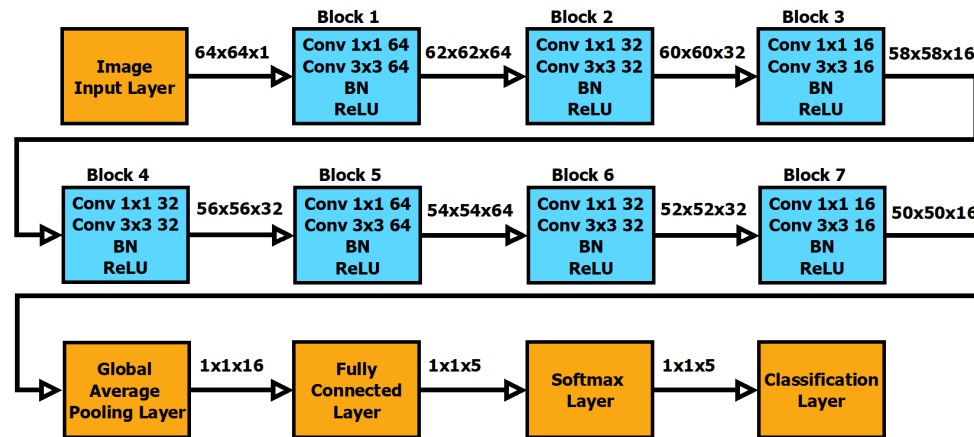

**Figure 5.** Block diagram of proposed CNN.

Each block in the network has two convolutional layers, followed by the batch normalization (BN) layer and the ReLU layer. No padding is used in any layer of the proposed CNN to retain the maximum statistical information. The first convolution layer performs 1x1 point-wise convolution. The second convolution layer performs $3 \times 3$ depth-wise convolution. In steganalysis [31], $1 \times 1$ point-wise convolution improves the results when applied with a depth-wise convolution. The training parameters of the proposed seven CNN blocks are 113,728. However, training parameters become 175,680 while considering a $3 \times 3$ size filter instead of a $1 \times 1$ size filter in the first convolutional layer in each block. The computational complexity can reduce by using a $1 \times 1$ filter and a $3 \times 3$ filter, consecutively. The performance improvement is also noticed in experiments as shown in Figure 6. The detection accuracy using Conv $1 \times 1$ is 92.47% and 91.37% for Conv 3 $\times$ 3 in the first layer of each block for two operator sequences, where $\alpha$ = GAU_1.0 & $\beta$ = MF5 with JPEG compression Q1 = 85 & Q2 = 75. The training time using Conv $1 \times 1$ is only one-third in comparison to Conv $3 \times 3$ training time and also has an improvement in the detection accuracy. Therefore, there are two considerable benefits of following the bottleneck approach.

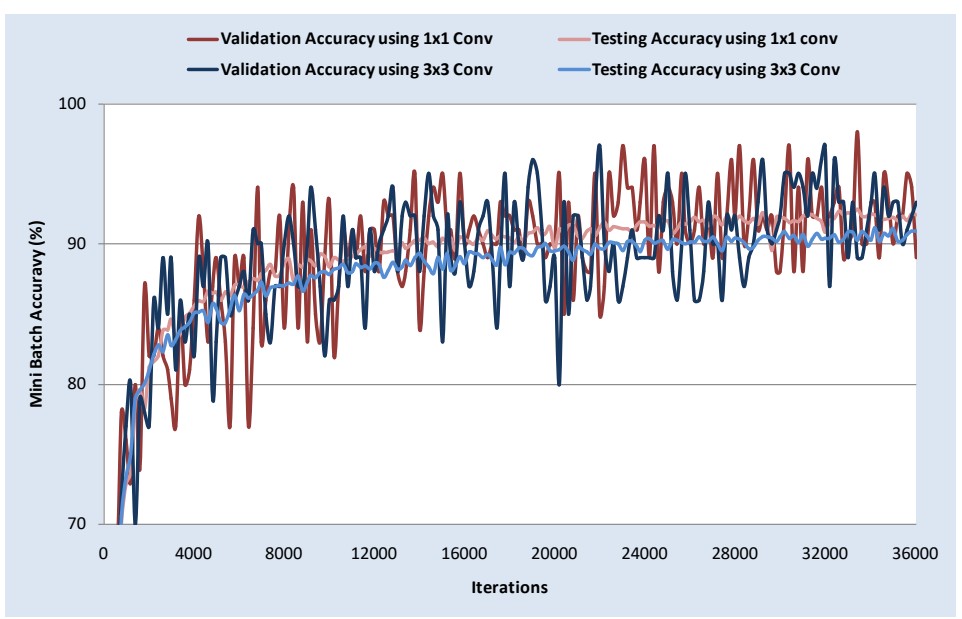

**Figure 6.** Analysis of Conv $1 \times 1$ vs. Conv $3 \times 3$.

The abstract diagram of the bottleneck approach and internal detail of blocks can be observed in Figure 7. The first and second convolution layer contains 64 filters of size $1 \times 1$ and $3 \times 3$ in blocks 1 and 5. Equal numbers of filters are considered as filter size $1 \times 1$ in $3 \times 3$ convolutional layers for every particular block. Convolutional layers are utilized 32 filters in block 2, block 4, and block 6. Convolutional layers are utilized 16 filters in block 3 and block 7. The stride of one is considered in each convolution layer. The batch normalization (BN) layer is used to increase the training pace and decrease the sensitivity to network initialization. The BN layer can diminish the inner covariant shift [32]. The learning parameters are updated according to the mean and variance of a mini-batch. After the training process is completed, the final values mean and variance of the BN layer are used for predicting the unseen data. The ReLU layer [33] replaces negative values with zero to improve the network performance.

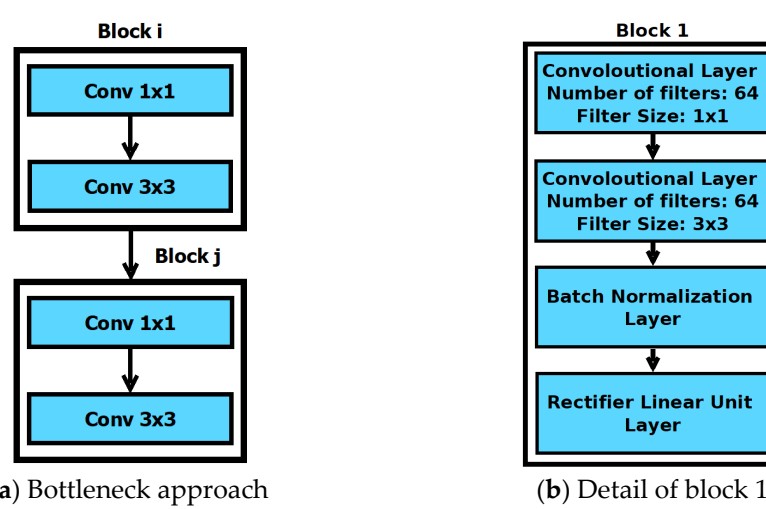

(**a**) Bottleneck approach  (**b**) Detail of block 1

**Figure 7.** Bottleneck approach and details of blocks.

Merely numbers of filters are changed in other blocks; the rest of the detail is similar to block 1. After the seven blocks, the global average pooling layer is followed by a fully connected layer, softmax layer, and classification layer. As the internal statistical information details are very crucial and the size of the image is also small, therefore only one global average pooling (GAP) layer is applied to prevent further information loss in the proposed network. In steganalysis [34,35], the global average pooling layer can enhance performance. Global average pooling is applied to achieve a single element from each feature map. The global average pooling layer increases the efficiency of the fully connected layer.

It is discovered in the experimental analysis that the GAP layer can increase the detection accuracy from 1% to 3%. The GAP is applied in the end only to retain the operation fingerprints. The GAP layer also reduces the overfitting issue [36]. The detection accuracy with the GAP layer is 92.47% and is 90.01% without the GAP layer for operator sequence GAU_1.0 and MF5 with JPEG compression Q1 = 85 & Q2 = 75. Therefore, there is a benefit of more than 2% in the detection accuracy after using the GAP layer. Additionally, as shown in Figure 8, the difference between validation and the testing accuracy is much less while using the GAP layer and without the GAP layer. The overfitting issue is well tackled by the GAP layer with a performance improvement.

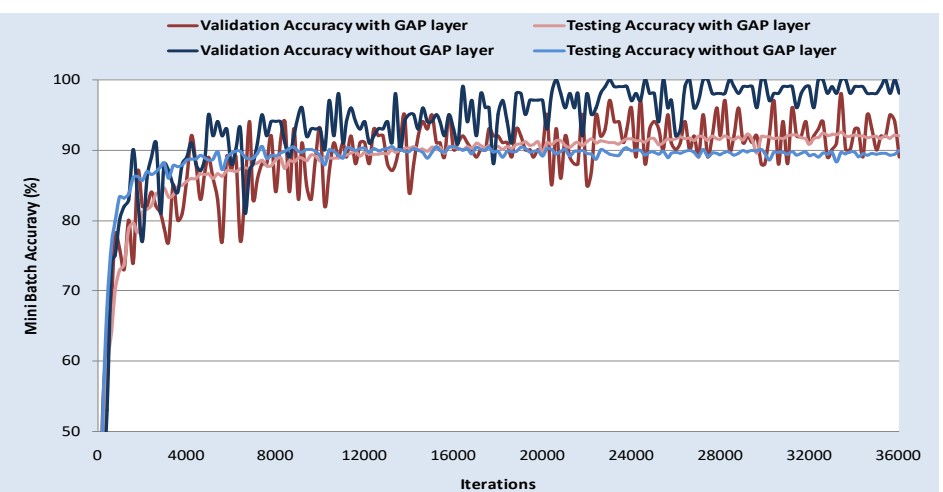

**Figure 8.** GAP layer effect on validation and testing accuracy.

Experiments are also performed with multiple pooling layers, however, they lead to poor performance in the end. Therefore, a single GAP layer is considered in the experiment analysis section. In the proposed CNN, the GAP layer produces 16 features, as the last convolutional layer has 16 filters. The fully connected layer combines all of the information learned from the previous layers. The input is multiplied by the weight matrix and the bias is added. The output size of the fully connected layer is five according to our problem, which has five classes. The output of the fully connected layer is processed by the softmax function. The softmax function is assigned the probability to every class. However, the sum of all probabilities should be 1. Finally, the classification layer assigns the exclusive class according to cross-entropy loss. Weight initialization of CNN is very crucial and can affect performance considerably. Additionally, random values are taken for network initialization in the previous step. However, it is not a practical solution and the performance of the network cannot be compared due to weight initialization. Glorot and Bengio [37] suggested a weight initialization strategy to give a better performance and fast convergence. The approach is more suitable for a less deep network like our proposed CNN. The weights are initialized according to the number of inputs and hidden nodes. The filter's behavior can be understood by analyzing as in Figure 9. In the first column, two pristine images are shown and their corresponding filtered images are shown in 4 × 4 tiles in the second and third columns. Second column images filtered from layer 6 kernels displayed the coarse details, and third column images filtered from layer 15 kernels showed the fine details. The behavior of layer kernels is changed according to the position of the layer. The information provided by layers kernels becomes coarse to fine while traversing the network from start to end.

The proposed network parameters are tuned with the help of exhaustive experimental analysis. The following parameters are considered in the network design. The stochastic gradient descent (SGD) algorithm is applied to curtail the loss function. In each iteration, mini-batch SGD is used to calculate the gradient and revise the weight and biases. The softmax classifier is utilized to minimize the cross-entropy between the estimated class and true class, where the momentum quantity is taken as 0.9, the number of epochs is 30, L2-regularization is 0.0004, and the initial learning rate is 0.001. The data is shuffled to avoid any unfairness toward unseen data in each epoch.

In the next section, experimental results are discussed. Most of the experiments are performed to detect the dual operator sequence. Experiments are performed together for uncompressed and JPEG compressed images.

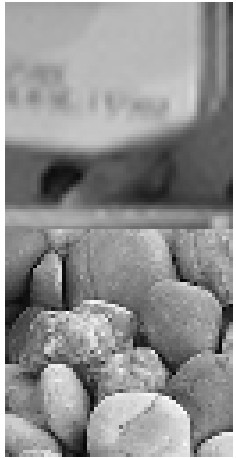 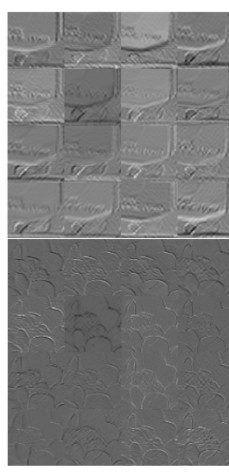 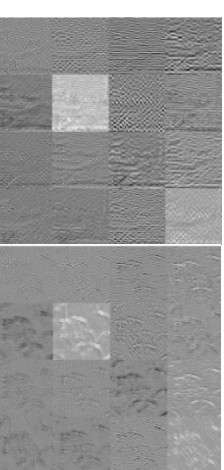

**Figure 9.** Filtered images of layer 6 and 15 kernels.

## 4. Experimental Results

Various experiments are performed to confirm the robustness and versatility of the proposed network. The first dataset is created using UCID [38], LIRMM [39], and Never-compressed (NC) [40] image databases, where UCID, LIRMM and NC contain 1338, 10,000 and 5150 uncompressed color images, respectively. The center block with $256 \times 256$ of each image is taken from databases. Further, 16 non-overlapping blocks of size $64 \times 64$ pixels are created. In the final, 263,808 image patches of size $64 \times 64$ pixels are generated and 30,000 patches of size $64 \times 64$ pixels are selected in each operator. Five operators—Gaussian blurring (GAU_P), median filtering of filter size $3 \times 3$ and $5 \times 5$ (MF3, MF5), un-sharp masking sharpening (SH_P), and up-sampling (UP_P) with different parameters *P*—are considered in experimental analysis. Symmetric padding is considered while applying the operator to the image. It is important to observe that 30,000 patches are selected randomly to get unbiased results for each operator. In the first dataset, 24,000 images are used for training and 6000 images for validation for each class. BOSSbase [30] dataset is considered for testing as a cross-database. It contains 10,000 uncompressed images of size $512 \times 512$. A similar approach is applied to create the image patches as in the first dataset and 160,000 image patches with $64 \times 64$ size are created. For each operator, 15,000 patches are used for testing. Experiments are performed using NVIDIA GTX1070 GPU with 24 GB RAM.

In the next part, the results of two operators with compressed and non-compressed images are discussed. Next, the results are shown for dissimilar parameters like compression, and the results are compared with some state-of-the-art methods.

### 4.1. Detection of Dual Operators Sequence for Similar Specification

In this section, the robustness of the proposed technique is discussed on two operator sequences under multiple parameters of operators. The specification is the same as used in CNN model training. The detection accuracy of the two operator sequences is given for the possible five classes as discussed in Section 2.1. $\Omega 0$ class denotes the pristine image, $\Omega 1$ class denotes images operated by operator $\alpha$, $\Omega 2$ class denotes images operated by operator $\beta$, $\Omega 3$ class denotes images first operated by operator $\alpha$ than operator $\beta$, and $\Omega 4$ class denotes images first operated by operator $\beta$ than operator $\alpha$. There are 15,000 images of the five classes in the testing set. It can be noticed from the Confusion matrix of operators, Gaussian blurring with standard deviation 1.0 (GAU_1.0) sequence, and median filtering of filter size $3 \times 3$ (MF3) as shown in Figure 10, where the proposed CNN network can classify two operator sequence images with good accuracy. The experimental results show that 991 GAU_1.0 operated images out of 15,000 GAU_1.0 operated images are misclassified in class $\Omega 4$, which is first operated by MF3 and then operated by GAU_1.0.

|   | Ω0 | Ω1 | Ω2 | Ω3 | Ω4 |
|---|---|---|---|---|---|
| Ω0 | 14927 | 28 | 36 | 4 | 5 |
| Ω1 | 6 | 13994 | 3 | 6 | 991 |
| Ω2 | 18 | 3 | 14857 | 121 | 1 |
| Ω3 | 4 | 65 | 185 | 14647 | 99 |
| Ω4 | 1 | 149 | 2 | 5 | 14843 |

**Figure 10.** Confusion matrix of GAU_1.0 and MF3 operator sequence.

The detailed results of the detection accuracy for different operator pairs are shown in Table 1. Gaussian blurring with standard deviation 0.7 (GAU_0.7), 1.0 (GAU_1.0), median filtering of filter size 3 × 3 (MF3), 5 × 5 (MF5), un-sharp masking sharpening with radius 2.0 (SH_2.0), 3.0 (SH_3.0), up-sampling with factor 1.2 (UP_1.2) and 1.5 (UP_1.5) operators are considered, where pairs of operators are represented in Table 1 for α and β. The average detection accuracy of five classes is more than 92% in all of the cases. The operator sequence with operator SH_2.0 or SH_3.0 is more challenging when compared with other operators, as shown in Table 1. The internal statistical information of an image is highly affected after applying un-sharp masking than other operators, as can be seen in Figure 4.

**Table 1.** Operator sequence detection with similar specifications.

| α = | GAU_1.0 | GAU_1.0 | GAU_1.0 | GAU_1.0 | GAU_1.0 | GAU_1.0 | GAU_0.7 | GAU_0.7 | GAU_0.7 | GAU_0.7 | GAU_0.7 | GAU_0.7 |
|---|---|---|---|---|---|---|---|---|---|---|---|---|
| β = | MF3 | MF5 | SH_2.0 | SH_3.0 | UP_1.2 | UP_1.5 | MF3 | MF5 | SH_2.0 | SH_3.0 | UP_1.2 | UP_1.5 |
| Ω0 | 99.51 | 99.45 | 95.47 | 98.06 | 99.77 | 99.79 | 98.61 | 98.59 | 94.64 | 93.41 | 99.33 | 99.17 |
| Ω1 | 93.29 | 98.45 | 91.16 | 95.84 | 82.97 | 94.90 | 98.15 | 99.07 | 94.97 | 94.47 | 94.40 | 99.59 |
| Ω2 | 99.05 | 94.39 | 93.08 | 90.41 | 99.13 | 99.57 | 94.02 | 82.37 | 90.87 | 91.47 | 95.23 | 95.53 |
| Ω3 | 97.65 | 97.71 | 99.74 | 99.90 | 99.85 | 99.89 | 94.30 | 92.33 | 98.43 | 96.99 | 99.36 | 99.31 |
| Ω4 | 98.95 | 99.07 | 81.94 | 83.79 | 93.67 | 96.41 | 99.66 | 99.78 | 86.56 | 90.13 | 99.86 | 99.86 |
| Average Accuracy | 97.69 | 97.81 | 92.28 | 93.60 | 95.07 | 98.11 | 96.95 | 94.43 | 93.09 | 93.30 | 97.63 | 98.69 |
| α = | MF3 | MF3 | MF3 | MF3 | MF5 | MF5 | MF5 | MF5 | SH_2.0 | SH_2.0 | SH_3.0 | SH_3.0 |
| β = | SH_2.0 | SH_3.0 | UP_1.2 | UP_1.5 | SH_2.0 | SH_3.0 | UP_1.2 | UP_1.5 | UP_1.2 | UP_1.5 | UP_1.2 | UP_1.5 |
| Ω0 | 94.97 | 94.49 | 85.03 | 93.01 | 88.33 | 92.24 | 71.71 | 83.12 | 93.41 | 96.06 | 98.29 | 96.51 |
| Ω1 | 99.43 | 99.30 | 99.82 | 99.87 | 99.45 | 98.76 | 99.65 | 99.69 | 95.95 | 94.34 | 90.56 | 92.81 |
| Ω2 | 92.91 | 92.95 | 98.68 | 98.81 | 93.15 | 93.57 | 99.65 | 99.57 | 87.76 | 95.05 | 91.37 | 83.06 |
| Ω3 | 96.77 | 97.13 | 99.69 | 99.57 | 91.99 | 95.56 | 99.67 | 99.79 | 94.37 | 78.05 | 90.59 | 95.59 |
| Ω4 | 85.36 | 87.63 | 91.06 | 98.83 | 79.06 | 85.27 | 80.23 | 87.37 | 99.91 | 99.80 | 99.76 | 99.00 |
| Average Accuracy | 93.89 | 94.30 | 94.85 | 98.02 | 90.40 | 93.08 | 90.18 | 93.91 | 94.28 | 92.66 | 94.11 | 93.39 |

As shown in Figure 11, the behavior of operators can also be visualized in the plot of mini-batch loss. The curve of α = GAU_0.7 & β = UP_1.2 becomes stable shortly with fewer peaks and valleys in the curve if compared with α = GAU_0.7 and β = SH_2.0. Similar attributes of stability are also followed by UP_1.5.

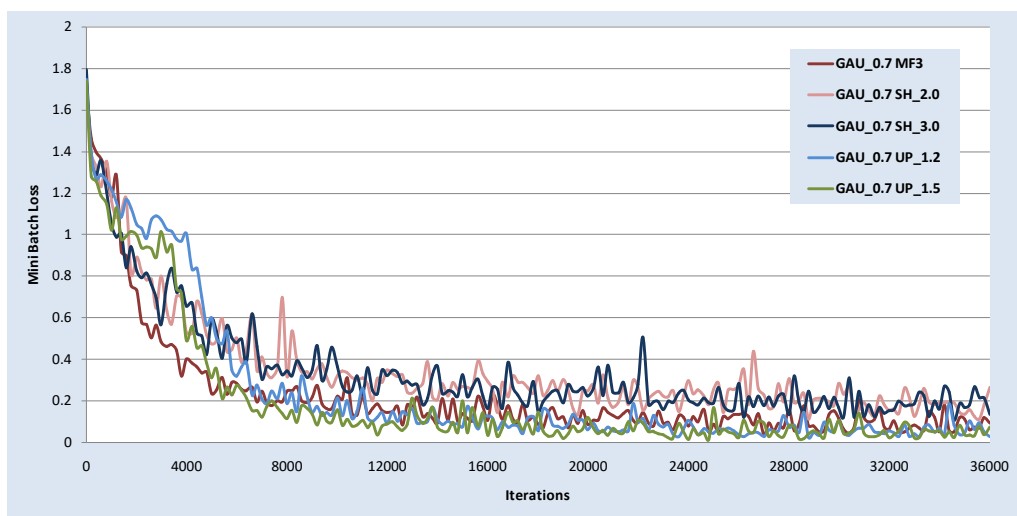

**Figure 11.** Mini-batch loss analysis.

The JPEG format is widely used when the visual quality remains good even after the compression in a real scenario as the default format. Therefore, three steps are considered in the detection of operator sequences in compressed images. In the first, the image is JPEG compressed with quality factor Q1. The operator sequence is applied to compressed images in step 2. In step 3, JPEG compression is applied with quality factor Q2. Detailed discussion regarding JPEG compression is given in Section 2.2. The confusion matrix of five class classifications on compressed images with Q1 = 75 and Q2 = 85 is shown in Figure 12. Two operators, Gaussian blurring with standard deviation 1.0 (GAU_1.0) and up-sampling with factor 1.5 (UP_1.5) are considered. The performance of the proposed CNN can degrade on compressed images in comparison to the uncompressed images. However, while considering the small image size ($64 \times 64$) and low compression quality factors, the performance is satisfactory.

|  | $\Omega0$ | $\Omega1$ | $\Omega2$ | $\Omega3$ | $\Omega4$ |
|---|---|---|---|---|---|
| $\Omega0$ | 14840 | 32 | 126 | 1 | 1 |
| $\Omega1$ | 18 | 12614 | 28 | 346 | 1994 |
| $\Omega2$ | 17 | 30 | 14816 | 76 | 61 |
| $\Omega3$ | 0 | 29 | 13 | 14640 | 318 |
| $\Omega4$ | 1 | 1897 | 18 | 2852 | 10232 |

**Figure 12.** Confusion matrix of GAU_1.0 and UP_1.5 operator sequence with Q1 = 75, Q2 = 85.

In Table 2, the results are shown for compressed images. The number of images for training, validation, and testing is similar to be considered in uncompressed images. Multiple compression quality factors are considered to visualize in the real scenario for Q1 = Q2, Q1 < Q2, and Q1 > Q2. The difference between quality factors Q1 and Q2 are also varied from 5 to 20. Compression omits the artifacts of the operator. Still, the average detection accuracy is nearly 90% in most of the cases. Even for $\alpha$ = GAU_0.8 and $\beta$ = MF3, the detection accuracy is more than 95%. Two cases are considered for $\alpha$ = GAU_1.0 and $\beta$ = MF5 on Q1 = 75 and Q2 = 85 as a first case and Q1 = 85 and Q2 = 75 as a second case. As can be seen in Table 2, the detection accuracy is 94.66 in the first case and 92.47

in the second. The high value of Q2 in the first case is the reason for its better detection accuracy in comparison to the second case. The artifacts of operators are better traceable for a high-quality factor compression.

**Table 2.** Operator sequence detection on compressed images with similar specifications.

| α | β | Compression | Ω0 | Ω1 | Ω2 | Ω3 | Ω4 | Average Accuracy |
|---|---|---|---|---|---|---|---|---|
| GAU_1.0 | MF5 | Q1 = 90, Q2 = 70 | 99.13 | 92.56 | 86.89 | 91.96 | 92.80 | 92.67 |
| GAU_1.0 | MF5 | Q1 = 75, Q2 = 85 | 98.91 | 94.27 | 88.14 | 92.85 | 99.13 | 94.66 |
| GAU_1.0 | MF5 | Q1 = 85, Q2 = 75 | 96.44 | 90.07 | 89.03 | 90.58 | 96.24 | 92.47 |
| GAU_1.0 | UP_1.5 | Q1 = 75, Q2 = 85 | 98.93 | 84.09 | 98.77 | 97.60 | 68.21 | 89.52 |
| GAU_1.0 | UP_1.5 | Q1 = 85, Q2 = 85 | 97.29 | 73.14 | 97.83 | 91.58 | 81.52 | 88.27 |
| GAU_0.9 | UP_1.2 | Q1 = 70, Q2 = 90 | 99.81 | 89.87 | 97.55 | 93.32 | 66.23 | 89.35 |
| GAU_0.8 | MF3 | Q1 = 70, Q2 = 90 | 99.59 | 95.76 | 94.67 | 95.72 | 91.45 | 95.44 |
| MF5 | UP_1.5 | Q1 = 75, Q2 = 85 | 97.28 | 78.22 | 98.35 | 96.29 | 79.21 | 89.87 |
| MF5 | UP_1.5 | Q1 = 85, Q2 = 75 | 97.39 | 76.70 | 97.97 | 96.79 | 71.89 | 88.15 |
| SH_2.0 | UP_1.2 | Q1 = 80, Q2 = 90 | 97,63 | 95.23 | 94,82 | 84.17 | 90.12 | 89.84 |
| SH_3.0 | UP_1.5 | Q1 = 80, Q2 = 90 | 98.21 | 96.97 | 94.56 | 83.67 | 89.45 | 92.57 |
| SH_3.0 | UP_1.5 | Q1 = 75, Q2 = 85 | 98.22 | 98.23 | 96.25 | 85.45 | 80.41 | 91.71 |

In this paper, all experiments are performed for the detection of two operator sequences except for single operator detection, as shown in Table 3. In Set 1, pristine images and images operated with four different operations—un-sharp masking, up-sampling, median filtering, and Gaussian blurring—are classified. The average detection accuracies for uncompressed images and compressed images with Q = 85 are 97.09% and 88.62%, respectively. Set 2 is constructed with the same operators as in Set 1, but with different parameter settings. In Set 2, the performance is also up to the mark.

**Table 3.** Single operator detection.

| Set | Single Operator | Uncompressed | Compression Q = 85 |
|---|---|---|---|
| Set 1 | ORI | 89.15 | 85.93 |
| | SH_2.0 | 97.01 | 82.14 |
| | UP_1.2 | 99.91 | 83.37 |
| | MF5 | 99.81 | 98.82 |
| | GAU_7.0 | 99.56 | 92.84 |
| | Average Accuracy | 97.09 | 88.62 |
| Set 2 | ORI | 97.39 | 90.81 |
| | SH_3.0 | 90.70 | 83.08 |
| | UP_1.5 | 99.99 | 90.92 |
| | MF3 | 99.81 | 95.11 |
| | GAU_1.0 | 99.93 | 95.17 |
| | Average Accuracy | 97.56 | 91.02 |

### 4.2. Detection of Dual Operators Sequence for Dissimilar Specification

Operators have the same parameter settings in training and testing for the experimental analysis above. However, operators may be the same, but the parameters of operators may vary in the real scenario. To assess the robustness of the proposed method against dissimilar specifications of operators, some experiments are performed. As can be seen

in the first row of Table 4, four values of standard deviation for Gaussian blurring {0.7, 0.8, 0.9, 1.0} are considered for training. A total of 60,000 images are used in training operated by Gaussian blurring and 15,000 images are operated by the four Gaussian blurring parameters. A total of 40,000 images are used to test operated by Gaussian blurring for 1334 images and operated by 300 Gaussian blurring parameters like the range of the parameters from 0.701 to 0.900. Therefore, a total of 300,000 images are used for training, and 200,000 are used for testing the detection of five-class classification problems of two operator sequence. Similarly, parameters are defined in Table 4 for other operators. The performance of the proposed CNN model is also excellent in dissimilar specifications. One scenario of compressed images with Q1 = 80 and Q2 = 90 is also displayed in Table 4 for operators α = GAU and β = UP. There is some reduction in the detection accuracy even still it is greater than 94%.

**Table 4.** Detection for dissimilar operator specifications.

| Operator | Training | Testing | Compression | Ω0 | Ω1 | Ω2 | Ω3 | Ω4 | Average Accuracy |
|---|---|---|---|---|---|---|---|---|---|
| α<br>β | GAU = {0.7, 0.8, 0.9, 1.0}<br>UP = {1.5, 1.6, 1.7, 1.8} | GAU = {0.701, 0.702, … , 0.899, 0.900}<br>UP = {1.500, 1.501, … , 1.799, 1.800} | No | 99.11 | 96.75 | 99.47 | 99.73 | 95.07 | 98.02 |
| α<br>β | GAU = {0.7, 0.8, 0.9, 1.0}<br>UP = {1.5, 1.6, 1.7, 1.8} | GAU = {0.701, 0.702, … , 0.899, 0.900}<br>UP = {1.500, 1.501, … , 1.799, 1.800} | Q1 = 80<br>Q2 = 90 | 99.39 | 96.49 | 86.06 | 94.16 | 96.35 | 94.49 |
| α<br>β | GAU = {0.7, 0.8, 0.9, 1.0}<br>MF3, MF5 | GAU = {0.701, 0.702, … , 0.899, 0.900}<br>MF3, MF5 | No | 99.59 | 88.32 | 98.72 | 98.52 | 96.63 | 96.36 |
| α<br>β | GAU_1.0<br>UP = {1.4, 1.5, … , 1.9} | GAU_1.0<br>UP = {1.400, 1.401, … , 1.899, 1.900} | No | 99.84 | 95.30 | 98.17 | 96.71 | 99.09 | 97.82 |

Here, some results are discussed for dissimilar compression factors on training and testing images. In the first row of Table 5, the JPEG compression quality factors for training images are Q1 = 85 and Q2 = 75 and Q1 = 80 and Q2 = 75 for testing images. The other operator parameters are the same for training and testing. In that case, there is still 92.40% detection accuracy. However, the performance deteriorates when the compression quality factor difference is more than 5. The performance of the proposed model is robust for quality factor difference 5 in training and testing images.

**Table 5.** Detection for dissimilar JPEG compression quality factors.

| Operator | | Compression | | Ω0 | Ω1 | Ω2 | Ω3 | Ω4 | Average Accuracy |
|---|---|---|---|---|---|---|---|---|---|
| α | β | Training | Testing | | | | | | |
| GAU_1.0 | MF5 | Q1 = 85, Q2 = 75 | Q1 = 80, Q2 = 75 | 94.73 | 91.17 | 89.25 | 90.44 | 96.41 | 92.40 |
| GAU_1.0 | MF5 | Q1 = 75, Q2 = 85 | Q1 = 85, Q2 = 75 | 94.26 | 76.25 | 87.84 | 56.56 | 84.90 | 79.96 |
| GAU_1.0 | MF5 | Q1 = 85, Q2 = 75 | Q1 = 90, Q2 = 75 | 97.22 | 90.55 | 89.04 | 88.31 | 91.59 | 91.34 |
| GAU_1.0 | UP_1.5 | Q1 = 85, Q2 = 85 | Q1 = 75, Q2 = 85 | 87.03 | 72.22 | 99.51 | 91.44 | 80.77 | 86.19 |
| GAU_1.0 | UP_1.5 | Q1 = 75, Q2 = 85 | Q1 = 85, Q2 = 85 | 91.95 | 79.01 | 92.78 | 97.96 | 65.57 | 85.45 |
| MF5 | UP_1.5 | Q1 = 85, Q2 = 75 | Q1 = 75, Q2 = 85 | 78.54 | 80.41 | 98.15 | 70.97 | 96.62 | 84.94 |
| SH_3.0 | UP_1.5 | Q1 = 80, Q2 = 90 | Q1 = 75, Q2 = 85 | 90.64 | 99.30 | 61.39 | 83.93 | 86.01 | 84.26 |
| SH_3.0 | UP_1.5 | Q1 = 75, Q2 = 85 | Q1 = 80, Q2 = 90 | 93.58 | 86.78 | 99.08 | 69.63 | 69.92 | 83.80 |

### 4.3. Comparative Analysis

The proposed CNN can classify two operators and their sequence in low resolution, compressed, and uncompressed images. The detailed experimental analysis is discussed above. Here, the results of the proposed scheme are compared with some other state-of-the-art techniques. In the CNN model [13], a constrained convolutional layer is introduced, unlike other traditional models. Different size filters, $7 \times 7$, $5 \times 5$, and $3 \times 3$ are used in the convolutional layer. In our experimental analysis, small-size filters are more suitable for detecting an image processing operation as shown in Figure 13. Bayar and Stamm [25] applied a modified constrained convolution layer on image residual for better results. The results are improved after modification, but still, there is a gap in the performance due

to a lower number of convolutional layers and large-size filters. Liao et al. [26] proposed the two-stream CNN model. The results of the two-stream model are impressive. The idea of detection of operator sequence is also a milestone in the research. The two-stream model can detect known as operators with unknown specifications. The computational cost is very high due to a large number of layers and customized preprocessing that need to be applied for the detection of different operators and compressed images. Our proposed network is moderate in size and requires less computation due to the bottleneck approach. The bottleneck approach can reduce the learning parameters and allow for increasing the network depth. Results are compared in Figure 13 for multiple scenarios. Both uncompressed and compressed types of images are considered in the comparison. The operators $\alpha$, $\beta$, and compression status are shown in the first, second, and third row in Figure 13.

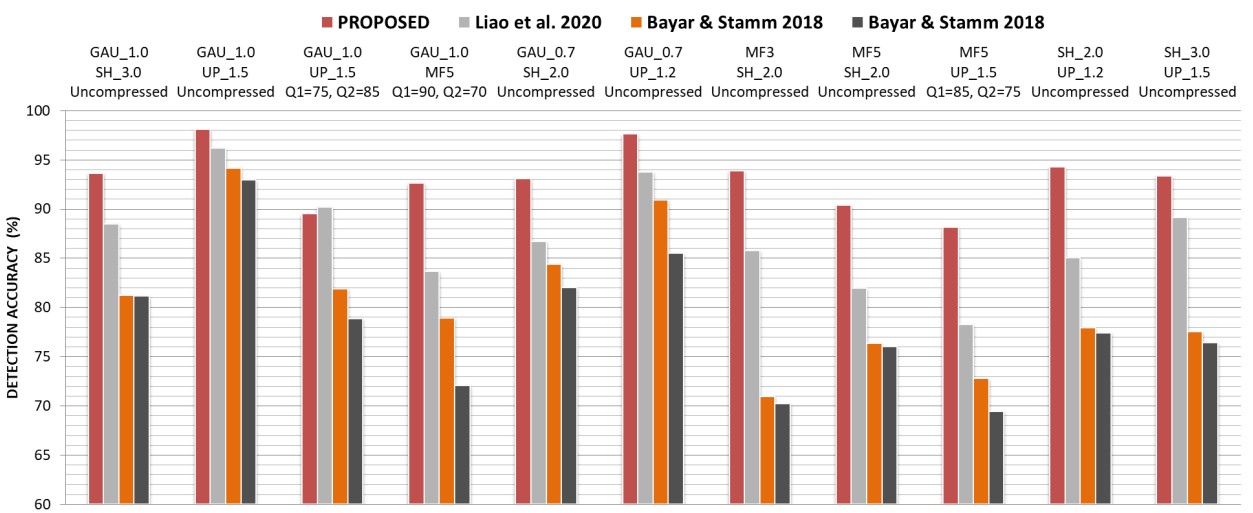

**Figure 13.** Comparative analysis in multiple scenarios.

Further, a detailed comparative analysis is performed with the method [26] because it is comparable with the proposed method. The other compared methods [13,25] performance is quite low as displayed in Figure 13. In Table 6, the average classification error of five class classifications is presented. The performance of Liao et al. [26] is inferior to our proposed architecture due to several reasons. Primarily, Liao's CNN contains multiple pooling layers that lose vital statistical information. Large size kernels have also reduced the performance. In the proposed CNN, a non-reducing approach is followed to retain inherited fingerprints as many as possible. The minimum reduction in classification error is for $\alpha$ = MF5 & $\beta$ = UP_1.5, i.e., 2.25% and the highest reduction in classification error is for $\alpha$ = SH_2.0 & $\beta$ = UP_1.2, i.e., 9.22% of the proposed scheme. Therefore, there is substantial improvement in detection performance.

In Table 7, a comparative analysis is given for JPEG compressed images. For uncompressed images, the performance of the proposed CNN is already found superior to Liao's CNN. Multiple compression factors are considered for unbiased analysis. The bottleneck approach allows fourteen convolution layers without the extra burden on computational cost. The classification error of Liao et al. [26] is less than the proposed scheme only in one case, i.e., $\alpha$ = GAU_1.0 & $\beta$ = UP_1.5. Otherwise, the proposed CNN outperforms with a good score. The average classification error of the proposed method is less and exclusive of specific preprocessing.

**Table 6.** Comparative analysis for uncompressed images.

| Operator | | Classification Error (%) | | Operator | | Classification Error (%) | |
|---|---|---|---|---|---|---|---|
| $\alpha$ | $\beta$ | Proposed | Liao et al. [26] | $\alpha$ | $\beta$ | Proposed | Liao et al. [26] |
| GAU_1.0 | MF3 | 02.31 | 07.39 | MF3 | SH_2.0 | 06.11 | 14.24 |
| GAU_1.0 | MF5 | 02.19 | 05.98 | MF3 | SH_3.0 | 05.70 | 13.81 |
| GAU_1.0 | SH_2.0 | 07.72 | 13.25 | MF3 | UP_1.2 | 05.15 | 10.83 |
| GAU_1.0 | SH_3.0 | 06.40 | 11.49 | MF3 | UP_1.5 | 01.98 | 07.49 |
| GAU_1.0 | UP_1.2 | 04.93 | 08.79 | MF5 | SH_2.0 | 09.60 | 18.02 |
| GAU_1.0 | UP_1.5 | 01.89 | 03.77 | MF5 | SH_3.0 | 06.92 | 15.49 |
| GAU_0.7 | MF3 | 03.05 | 08.26 | MF5 | UP_1.2 | 09.82 | 13.37 |
| GAU_0.7 | MF5 | 05.57 | 08.01 | MF5 | UP_1.5 | 06.09 | 08.34 |
| GAU_0.7 | SH_2.0 | 06.91 | 13.31 | SH_2.0 | UP_1.2 | 05.72 | 14.94 |
| GAU_0.7 | SH_3.0 | 06.70 | 12.70 | SH_2.0 | UP_1.5 | 07.34 | 11.67 |
| GAU_0.7 | UP_1.2 | 02.37 | 06.23 | SH_3.0 | UP_1.2 | 05.89 | 13.54 |
| GAU_0.7 | UP_1.5 | 01.31 | 05.95 | SH_3.0 | UP_1.5 | 06.61 | 10.84 |

**Table 7.** Comparative analysis of compressed images.

| Operator | | Compression | Classification Error (%) | |
|---|---|---|---|---|
| $\alpha$ | $\beta$ | | Proposed | Liao et al. [26] |
| GAU_1.0 | UP_1.5 | Q1 = 75, Q2 = 85 | 10.48 | **09.80** |
| GAU_1.0 | UP_1.5 | Q1 = 85, Q2 = 85 | **11.73** | 14.68 |
| GAU_1.0 | MF5 | Q1 = 90, Q2 = 70 | **07.33** | 16.35 |
| GAU_1.0 | MF5 | Q1 = 75, Q2 = 85 | **05.34** | 11.82 |
| GAU_1.0 | MF5 | Q1 = 85, Q2 = 75 | **07.53** | 15.93 |
| GAU_0.9 | UP_1.2 | Q1 = 70, Q2 = 90 | **10.65** | 14.12 |
| MF5 | UP_1.5 | Q1 = 75, Q2 = 85 | **10.13** | 13.25 |
| MF5 | UP_1.5 | Q1 = 85, Q2 = 75 | **11.85** | 21.75 |
| MF3 | GAU_0.8 | Q1 = 70, Q2 = 90 | **04.56** | 12.70 |
| SH_2.0 | UP_1.2 | Q1 = 80, Q2 = 90 | **10.16** | 14.65 |
| SH_3.0 | UP_1.5 | Q1 = 80, Q2 = 90 | **07.43** | 13.55 |

## 5. Conclusions

Digital images have turned out to be the most accepted representation of information. The latest technologies have empowered naive users to create fake images effortlessly. Though, multiple operations have been performed to construct a real-looking fake image. The detection of manipulation operations has assisted to find the fake image. At present, deep learning approaches have been taken into the place of handcrafted feature extraction methods. A convolutional neural network has been suggested to preserve the authenticity of images. Thus, most techniques have been suggested for single operator detection so far. Very few techniques have been discussed to identify the dual operators and the order of operators. The proposed deep learning model could detect consecutive dual operators on the image and its corresponding order precisely. The bottleneck approach has been applied in the model to increase the layers and reduced the parameters. Unlike previous networks, one single global averaging pooling layer has been utilized to reduce the information loss and overfitting problem. The proposed model has been performed

robustly against challenging scenarios like low-resolution images and compression in the exhaustive experimental analysis.

**Author Contributions:** Each author discussed the details of the manuscript and contributed equally to its preparation. S.-H.C. designed and wrote the manuscript. S.A. implemented the proposed technique and provided the experimental results. S.-J.K. reviewed the article. K.-H.J. drafted and revised the manuscript. All authors have read and agreed to the published version of the manuscript.

**Funding:** This research was supported by Basic Science Research Program through the National Research Foundation of Korea (NRF) funded by the Ministry of Education (2021R1I1A3049788) and Brain Pool program funded by the Ministry of Science and ICT through the National Research Foundation of Korea (2019H1D3A1A01101687, 2021H1D3A2A01099390).

**Institutional Review Board Statement:** Not applicable.

**Informed Consent Statement:** Not applicable.

**Data Availability Statement:** The datasets used in this paper are publicly available and their links are provided in the reference section.

**Acknowledgments:** We thank the anonymous reviewers for their valuable suggestions that improved the quality of this article.

**Conflicts of Interest:** The authors declare no conflict of interest.

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
