# Peer review of "Image Forensics Using Non-Reducing Convolutional Neural Network for Consecutive Dual Operators"

_applsci, doi:10.3390/app12147152_

Round 1
Reviewer 1 Report
This paper presents a robust moderate-sized convolutional neural network to identify manipulation operators and sequences to make the network deeper reduce the computational cost utilized to retain the maximum flow of information and to avoid the overfitting issue between layers. The proposed network is claimed to be robust against the low resolution and JPEG compressed images with challenging limited availability of statistical information. Interestingly, the proposed model is to detect operator with different parameters and compression quality factors that are not considered in training making it a promising research. The work is almost complete needing to involve some matters to be ready. In this regard, the author is requested to consider the following topics: novel embedding secrecy within images utilizing an improved interpolation-based reversible data hiding scheme, improving data hiding within colour images using hue component of HSV colour space, efficient implementation of multi-image secret hiding based on LSB and DWT steganography comparisons, efficient image reversible data hiding technique based on interpolation optimization, boosting image watermarking authenticity spreading secrecy from counting‐based secret‐sharing, refining image steganography distribution for higher security multimedia counting-based secret-sharing, efficient reversible data hiding multimedia technique based on smart image interpolation, hiding shares by multimedia image steganography for optimized counting-based secret sharing, watermarking images via counting-based secret sharing for lightweight semi-complete authentication.
Author Response
Thank you for your kind review and honorable comments.
Please check the attached file.

Reviewer 2 Report
Submitted article discusses an interesting and up-to-date problem of detection of manipulations carried out on images with the use of modern technologies. The so-called fake images or deep fakes i.e. fakes produced with the use of deep learning algorithms are used to mislead people in many areas of life.
The authors described in details the architecture and the idea of proposed algorithm of Convolutional Neural Network dedicated to classify a type of modification caried out on an image and what more an order of modifications in the case of two modifications and presented results of experiments conducted for many different scenarios.
The article has been prepared with great care, and the presented research results are clear, however, there are some issues that should be considered for modification or completion. Details were described below:
1) In lines 45-47, the statement „ In the recent era of a deep learning network, a convolutional neural network has given propitious results, especially in object classification, face recognition, medical imaging, and so on.” should be confirmed with a few examples with references – there are a lot.
2) The article should be checked for typos, below is one of the found:
a) Figure 7b – Rctifier instead of Rectifier Linear Unit Layer/
3) In Figures 6th and 8th authors presents the comparison of validation and testing curves for two variants of their algorithm. Testing is a procedure carried out on a test dataset with the usage of the trained algorithm, so testing results should be presented should present for only one test iteration. The curves presented by the authors look more like training and validation curves.
4) In Tables 6 and 7 there is no information on the evaluation metric applied to compare algorithms. This information there is in the text of the article, but the presentation of results will be more clear with this information but also in the table header.
Author Response

(The authors gave the same response as above.)

Reviewer 3 Report
This paper designs a network for image forensics on the order of operations. The paper is well organized and can be accepted with minor revision. My comments are as follows:
- The relationship between the proposed method and the formulated problem in section 2.1 should be discussed. In other words, which part of the proposed method is designed for the detection on consecutive dual operators.
- The first paragraph of Introduction is too long. A individual paragraph of the summary of existing schemes about image forensics is necessary.
- In Fig. 13, the authors compared their network with related work [13], [25], and [26]. However, in Table 6 and 7, only the scheme in [26] was employed for comparison. The reasons should be explained.
Author Response

(The authors gave the same response as above.)
